# Reelin Haploinsufficiency and Late-Adolescent Corticosterone Treatment Induce Long-Lasting and Female-Specific Molecular Changes in the Dorsal Hippocampus

**DOI:** 10.3390/brainsci8070118

**Published:** 2018-06-25

**Authors:** Anna Schroeder, Maarten van den Buuse, Rachel A. Hill

**Affiliations:** 1The Florey Institute of Neuroscience and Mental Health, University of Melbourne, Parkville 3052, Australia; anna.schroeder@monash.edu; 2Department of Psychiatry, School of Clinical Sciences, Monash University, Clayton 3168, Australia; 3School of Psychology and Public Health, La Trobe University, Bundoora 3086 Australia; m.vandenbuuse@latrobe.edu.au; 4Department of Pharmacology, University of Melbourne, Parkville 3052, Australia; 5The College of Public Health, Medical and Veterinary Sciences, James Cook University, Townsville 4810, Australia

**Keywords:** reelin, corticosterone, glucocorticoid receptors, NMDA receptors, parvalbumin, GAD_67_

## Abstract

Reelin depletion and stress seem to affect similar pathways including GABAergic and glutamatergic signaling and both are implicated in psychiatric disorders in late adolescence/early adulthood. The interaction between reelin depletion and stress, however, remains unclear. To investigate this, male and female heterozygous reelin mice (HRM) and wildtype (WT) controls were treated with the stress hormone, corticosterone (CORT), during late adolescence to simulate chronic stress. Glucocorticoid receptors (GR), *N*-methyl-d-aspartate receptor (NMDAr) subunits, glutamic acid decarboxylase (GAD_67_) and parvalbumin (PV) were measured in the hippocampus and the prefrontal cortex (PFC) in adulthood. While no changes were seen in male mice, female HRM showed a significant reduction in GR expression in the dorsal hippocampus. In addition, CORT reduced GR levels as well as GluN2B and GluN2C subunits of NMDAr in the dorsal hippocampus in female mice only. CORT furthermore reduced GluN1 levels in the PFC of female mice. The combined effect of HRM and CORT treatment appeared to be additive in terms of GR expression in the dorsal hippocampus. Female-specific CORT-induced changes were associated with overall higher circulating CORT levels in female compared to male mice. This study shows differential effects of reelin depletion and CORT treatment on GR and NMDAr protein expression in male and female mice, suggesting that females are more susceptible to reelin haploinsufficiency as well as late-adolescent stress. These findings shed more light on female-specific vulnerability to stress and have implications for stress-associated mental illnesses with a female bias including anxiety and major depression.

## 1. Introduction

Reelin is an extracellular matrix protein that is secreted by Cajal-Retzius cells during embryonic brain development and mainly by cortical and hippocampal GABAergic interneurons during adulthood [1,2,3]. During brain development, reelin is predominantly involved in cortical layer formation [1]. Mice with homozygous loss of reelin, called reeler mice, display inversion of cortex cell layers and malposition of neurons throughout the hippocampus [4]. Heterozygous loss of reelin in mice induces a less severe phenotype showing dendritic abnormalities in the cortex and the hippocampus and only subtle abnormalities in cognitive function [5,6,7,8,9]. These mice are used to investigate the role of reelin in the development of psychiatric disorders as mutations in the reelin gene as well as decreased levels of mRNA are associated with autism [10], schizophrenia and bipolar disorder [2,11].

Aside from its developmental role, reelin serves a crucial function in the mature brain acting through its two major receptors, Apolipoprotein E receptor 2 (EpoER2) and very low-density lipoprotein receptor (VLDLR) [12], activating numerous downstream cascades via Dab1 phosphorylation such as Akt/mTOR or N-WASP leading to microtubule formation and actin stabilization, respectively [13]. In the mature brain, reelin signaling promotes dendritic spine maturation, synaptogenesis, synaptic transmission and plasticity, and therefore plays a critical role in the modulation of synaptic circuits [14,15,16]. In a comprehensive review of articles published from 1980 to 2014 on major depression a main finding was the association between both chronic stress and major depression with dendritic spine loss and aborization as well as reduced glial cells [17], thus disruption to reelin may be upstream of this broad set of functional deficits associated with major depression. 

Reelin signaling through ApoER2 was shown to activate NMDAR phosphorylation and calcium influx through these receptors and subunit composition [18] thus mediating synaptic plasticity and learning and memory [14,16].

NMDAr are tetramers consisting of two GluN1 subunits and two GluN2 subunits (GluN2A, GluN2B, GluN2C and GluB2D). While NMDA receptors containing GluN2C and GluN2D are mainly localized to the cerebellum and show slower kinetics compared to GluN2A and GluN2B in the adult brain [19], GluN2A and GluN2B are highly expressed in the hippocampus and the cortex and are critical mediators of synaptic function via regulating synaptic plasticity (reviewed in Sanz-Clemente, Nicoll [20]). Reelin has also been closely linked to the markers of GABAergic interneurons: glutamic acid decarboxylase (GAD_67_), the enzyme that converts glutamate to GABA, and parvalbumin (PV), as these were simultaneously decreased with reelin mRNA and protein levels in the PFC and the hippocampus in schizophrenia and bipolar post-mortem brains [21] as well as in mouse models of these neuropsychiatric disorders [22,23]. Findings on the effect of reelin deficiency on the expression of these GABAergic markers as well as NMDAr subunits remain inconsistent, possibly due to a lack of stratification for sex. Nullmeier et al. [23] for example combined male and female data when comparing GAD_67_ expression in reelin haploinsufficient and wild-type mice, and found a significant decrease in GAD_67_ and PV in multiple hippocampal regions, but did not stratify for sex in the analysis. Previously, we have shown sex differences in BDNF, TrkB and GluNR2C expression in reelin haploinsufficient mice [24,25], while others have shown that male but not female Rln +/− mice show alterations in steroid hormone levels in the cerebellum when compared to controls [26]. Due to the well-established sex differences within GABAergic development [27] as well as psychiatric disorders [28,29], it is important to discern specific differences between males and females.

While several genetic variants have been associated with severe psychiatric disorders such as major depression, no single gene is likely to be causative. Rather a gene × gene or gene × environment interaction is more probable. A well-accepted model to explain the complex and heterogeneous aetiology of psychiatric disorders is the two-hit hypothesis, which suggests that early genetic or environmental insults disrupt the developing brain causing it to be vulnerable to a second environmental insult during adolescence, such as stress [30]. Reelin deficiency may constitute a risk factor by making the brain more vulnerable to stress. In particular adolescent stress has been shown to be a major environmental trigger for psychiatric illnesses as the brain still undergoes synaptic changes and maturation, including of hypothalamic-pituitary-adrenal (HPA) axis activity [31], during this vulnerable period of development. Chronic stress is well known for its detrimental effects on neuronal morphology and function in brain regions such as the hippocampus and PFC [32,33] as these show the highest density of the glucocorticoid receptors (GR), which bind the stress hormone cortisol (in humans) and corticosterone (in rodents) [34]. In particular GABAergic interneurons as well as NMDA receptors appear to be affected by stress [33,35,36], targets that are also vulnerable to reelin deficiency. While there is strong evidence that reelin deficiency and stress may affect common pathways, only a few studies looked at the interaction underlying this genetic risk factor and the environmental insult mainly focusing on the behavioral phenotype related to neuropsychiatric disorders [37,38]. We have previously shown that heterozygous reelin mice (HRM) were more susceptible to CORT treatment with regards to spatial memory [37], however, it remains unclear how reelin deficiency may increase susceptibility to stress on the molecular level. This study sought to investigate a) how reelin deficiency or adolescent stress impact the expression of NMDA receptor subunits, the GABAergic markers, GAD_67_ and PV, as well as GR and b) whether reelin deficiency makes the brain more vulnerable to adolescent/early adult stress which may explain the profound decrease in those markers in psychiatric disorders. Given strong evidence for sex differences in HPA axis activity and responses to stress [39] as well as in the incidence of psychiatric disorders such as major depression [28] and schizophrenia [29], we assessed both male and female mice. We used HRM which show 50% reelin depletion [24]. Late-adolescent stress was simulated by chronic corticosterone (CORT) administration [40]. We analyzed the PFC and hippocampus, brain regions which are susceptible to stress and display high levels of reelin and GR expression [34,41]. We hypothesized that the combined effect of reelin haploinsufficiency and adolescent CORT treatment would cause alterations in GABAergic, glutamatergic and stress receptor pathways but these may be differentially modulated according to sex.

## 2. Materials and Methods

### 2.1. Animals

Mice were derived from a breeding colony at the Florey Institute of Neuroscience and Mental Health, Melbourne, Australia, and consisted of both male and female HRM and wild-type (WT) control mice. The colony was originally established with Reln^rl/+^ (on a C57Bl/6J genetic background) and crossed with C57Bl/6J breeders that were purchased from The Jackson Laboratory (Bar Harbor, ME, USA). All mice were group-housed in individually-ventilated cages (IVC, Tecniplast, Buguggiate, Italy) with *ad libitum* access to food and water, and kept on a 12/12 h light/dark cycle (lights on 7 a.m.). Cages were cleaned once a week. All procedures were conducted according to the guidelines in the Australian Code of Practice for the Care and Use of Animals for Scientific Purposes (National Health and Medical Research Council of Australia, *8th edition 2013*) and approved by the Animal Ethics Committee of the Florey Institute of Neuroscience and Mental Health.

### 2.2. Corticosterone Treatment

Mice received chronic CORT treatment in their drinking water [42,43] for 21 days starting from six weeks of age until nine weeks. We based the treatment time window on our previous developmental studies in C57Bl/6 mice, which showed a rapid rise in seminal vesicle weight, serum testosterone and uterine weight around 6–9 weeks of age, which we accordingly describe as the late-adolescent period [44]. CORT (Sigma-Aldrich, St. Louis, MO, USA) was dissolved in 100% ethanol and diluted with tap water to a final concentration of 50 mg/L (0.5% ethanol). We used CORT treatment rather than other stress paradigms in order to focus on glucocorticoid effects and its interaction with reelin signaling. Stress paradigms (e.g., chronic restraint stress) elicit many physiological responses and many broad molecular changes that differ compared to glucocorticoid-specific effects, making interpretation of the data more complex. Control mice received a vehicle solution (0.5% ethanol in tap water). After the three-week treatment period all mice received normal tap water and were left undisturbed until 15 weeks of age, when the brains and adrenal glands were collected for molecular analyses to detect long-term effects of CORT and/or reelin haploinsufficiency. Experimental groups consisted of four male and four female groups: WT mice treated during adolescence with vehicle (WT Contr); WT mice treated during adolescence with CORT (WT CORT); HRM treated during adolescence with vehicle (HRM Contr) and HRM treated during adolescence with CORT (HRM CORT) (*n* = 7). Body weight was recorded weekly from the beginning of the treatment.

### 2.3. Detection of CORT

To detect CORT levels during treatment, without causing adverse stress from extracting blood, we instead collected faecal boli. Samples were collected from their home cage at three time points: week 6 (prior to the beginning of CORT treatment); week 9 (at the completion of the treatment period); and week 11 (two weeks following cessation of treatment). The faecal boli were frozen in liquid nitrogen and crushed using a pestle and mortar. One ml of methanol was added to 50 mg samples and vortexed for 30 min, before collection of the supernatant. To detect CORT levels we used an enzyme-immunoassay kit (Cayman Chemical Company, Ann Arbor, MI, USA). With samples used at a 1:50 dilution. Touma, Palme [45], have previously detected CORT levels in the faeces of male and female rodents and report this technique as a robust measure of pharmacological stimulation and suppression of adrenocortical activity. Accurate representation of CORT by means of faecal CORT measurements was also supported by other studies [46,47].

### 2.4. Western Blot Analysis

Mice were killed by cervical dislocation at 15 weeks of age and their adrenal glands and brains were collected and stored at −80 °C. The prefrontal cortex and hippocampus were dissected and the hippocampus was separated into ventral and dorsal hippocampus (approximately 50/50). All dissections were performed by the same researcher to ensure consistent dissection ratios. Protein extraction and Western blot analysis were performed as previously described by Klug, Hill [40] and Buret and van den Buuse [24]. The primary antibodies were anti-glucocorticoid receptor (GR) (97 kDa, 1:500, ab2768, Abcam), anti-NMDAr2C (1:500, ab110, Abcam) which recognizes 180 kDa GluN2B, 140 kDa GluN2C and 120 kDa GluN1 subunits, anti-NMDAr2A (170 kDa, 1:1000, ab14596, Abcam), anti-parvalbumin (12 kDa, 1:2000, MAB1572, Millipore), and anti-GAD_67_ (67 kDa, 1:1000, Sigma-Aldrich).

### 2.5. Statistical Analysis

Data are expressed as mean ± standard error of the mean (SEM). Group differences were assessed with SYSTAT 13 (Systat Software Inc., San Jose, CA, USA). Five to six mice per group were used to analyze CORT levels as well as all presented protein levels. Relative adrenal weights as well as all Western blot results were analysed using three-way analysis of variance (ANOVA) with sex, genotype and (CORT) treatment as between-group factors. Faecal CORT levels were analyzed by means of repeated measures ANOVA with the same between-factors and time as a within-group factor. In all cases, the significance level was set to *p* ≤ 0.05.

## 3. Results

### 3.1. Adrenal Weight

Statistical analysis of relative adrenal weight at 15 weeks of age revealed a main treatment effect (F(1,82) = 9.5, *p* = 0.003) and a sex effect (F(1,82) = 271.0, *p* < 0.001). This reflected that the adrenal glands were considerably larger in females than males and that CORT-treated mice had smaller adrenals compared to their non-treated controls, irrespective of sex or genotype (Table 1). Body weight was smaller in females compared to males as expected and no differences were detected between the CORT and HRM groups at week 15 (data not shown).

### 3.2. CORT Levels

Repeated measures ANOVA of faecal CORT revealed a main effect of sex (F(1,26) = 19.3, *p* < 0.001), reflecting three-fold higher CORT levels in female compared to male mice, as well as a day × CORT interaction (F(2,52) = 6.7, *p* = 0.003) (Figure 1). Further analysis separated by time points of measurement (days: day 0—before CORT treatment, day 21—last day of CORT treatment, day 35—2 weeks after cessation of CORT treatment) showed a main sex effect for all three days (day 0: F(1,31) = 27.4, *p* < 0.001; day 21: F(1,33) = 5.7, *p* = 0.022; day 35: F(1,30) = 34.2, *p* < 0.001). No treatment or genotype differences were observed on days 0 and 35, while a main treatment effect (F(1,33) = 6.3, *p* = 0.018) was detected on day 21 reflected higher CORT levels in CORT-treated male and female groups versus vehicle-treated controls (Figure 1).

### 3.3. GR Expression

Statistical analysis of GR protein expression in the dorsal hippocampus (Figure 2A) revealed a significant sex × CORT interaction (F(1,39) = 3.05; *p* = 0.01). Further separate analyses of males and females showed a main CORT effect (F(1,19) = 7.37; *p* < 0.05) and a main genotype effect (F(1,19) = 7.02; *p* < 0.05) in females only, reflecting reduced levels of GR in the CORT-treated females as well as in HRM as compared to their respective control groups (Figure 2A). While no significant genotype × CORT interaction was found in females, the HRM + CORT group showed the lowest expression levels of GR. No significant differences in GR expression were detected in the dorsal hippocampus of males. No effects of genotype or CORT on GR expression were detected in the ventral hippocampus or PFC of either male or female mice.

### 3.4. NMDAr Protein Expression

#### 3.4.1. NMDAr Subunit Protein Expression in the Dorsal Hippocampus (Figure 3A–D)

While CORT had no effect on NMDAr expression in the dorsal hippocampus of male mice, it significantly reduced GluN2B and GluN2C subunit levels in female mice (Figure 3C,D). This was supported by an overall sex × CORT interaction (GluN2B: F(1,39) = 4.62; *p* < 0.05) GluN2C: F(1,39) = 9.1; *p* = 0.004) and further main CORT effects when data from female mice were analyzed separately (GluN2B: F(1,20) = 4.81; *p* = 0.04, GluN2C: F(1,20) = 11.15; *p* = 0.003). For GluN2C expression, a main sex effect was seen in vehicle-treated mice (F(1,20 = 12.47; *p* = 0.002) reflecting higher levels of GluN2C in females compared to male mice (Figure 3D). No differences were detected for GluN1 and GluN2A between the groups (Figure 3A,B).

#### 3.4.2. NMDAr Protein Expression in Ventral Hippocampus (Figure 3E–H)

Protein expression of GluN2C was lower in females as compared to male mice in the ventral hippocampus as shown by the main sex effect (F(1,34) = 9.20; *p* = 0.005) (Figure 3H). No significant effects of CORT, genotype or genotype × CORT interactions were observed in the ventral hippocampus for any of the subunits (Figure 3E–H).

#### 3.4.3. NMDAr Protein Expression in PFC (Figure 3I–L)

With regards to GluN1 expression in the PFC (Figure 3I), statistical analysis revealed a significant sex × CORT interaction (F(1,40 = 7.01; *p* < 0.05). Further analysis showed that CORT significantly reduced GluN1 protein expression in female mice (F(1,20) = 6.11; *p* < 0.05), while no significant differences were observed in males. Within the vehicle-treated groups a significant main sex effect was observed (F(1,20) = 5.53; *p* = 0.029) reflecting lower GluN1 levels in males compared to females (Figure 3I). No significant statistical differences were seen for the subunits GluN2A, GluN2B or GluN2C in the PFC (Figure 3J–L).

No significant effect of genotype and no genotype × CORT interaction was found for any NMDAr subtypes in any of the 3 brain regions.

### 3.5. PV and GAD_67_ Protein Expression

No significant effects of CORT or genotype and no genotype × CORT interactions were detected on PV or GAD_67_ protein expression in the dorsal hippocampus, ventral hippocampus or the PFC (Figure 4).

## 4. Discussion

The two main findings of this study are that (a) female, but not male HRM had lower GR levels in the dorsal hippocampus; and (b) stress as mimicked by CORT treatment reduced GR, GluN2B and GluN2C expression in the dorsal hippocampus of female, but not male mice. In the PFC, CORT reduced GluN1 expression in female, but not male mice. Although animals with reelin depletion did not show increased susceptibility to CORT treatment in this study, we demonstrated a female-specific genotype as well as CORT effects on GR and NMDA receptor expression. Furthermore, the combined effect of both reelin haploinsufficiency and CORT treatment was additive in terms of GR expression in the dorsal hippocampus, with this group showing the lowest expression level.

To ensure that CORT treatment specifically affected the late adolescent period, we measured CORT metabolites from mouse faeces on week 6 (start of CORT treatment), week 9 (end of CORT treatment) and week 11. As expected, CORT was elevated solely during the treatment period (6–9 weeks) as was shown by significantly higher CORT levels in CORT-treated animals compared to controls on the last day of the treatment and return to baseline levels two weeks after treatment cessation. Reduced reelin levels did not affect the levels of CORT as no differences were seen between the genotypes. No differences were seen between baseline CORT levels and the levels measured two weeks after CORT cessation reflecting normal adrenal function after the treatment period, although adrenal glands were significantly smaller in CORT-treated groups compared to controls on week 15. A striking finding was that female mice had approximately three-fold higher baseline-CORT levels compared to male mice and, accordingly, these levels were three times higher during CORT treatment. Consistent with our study, Touma et al. [45] reported that females generally showed values about twice as high as males. Although it is difficult to directly compare faecal and plasma CORT levels, as females show fluctuations in CORT plasma levels during the estrous cycle [48] and have higher plasma levels of corticosteroid-binding globulin (CBG) [49], a large number of studies demonstrate higher plasma CORT levels in females as compared to males [50,51,52,53]. An explanation for lower CORT metabolite expression in males might be the protective role of testosterone. A large number of studies have shown that androgens inhibit the HPA axis. Basal CORT levels are increased in male rats after gonadectomy (GDX) [51,54,55]. Blockade of the androgen receptor in adult male rats prior to restraint stress resulted in elevated CORT and ACTH responses [56,57]. Concomitantly, the female hormone, 17β-estradiol (E2), seems to maintain CORT at higher levels. Several studies reported decreased CORT levels as well adrenal weight after ovariectomy, which were reversed by E2 treatment [58,59]. In agreement with this observation and our results, Weathington, Arnold [60] showed that female rats were more severely affected by juvenile stress compared to males and showed higher circulating CORT levels.

The 3-fold higher level of CORT in the females compared to males in our study may explain the female-specific reduction in NMDAr subunits as well as GR expression. CORT reduced GR protein levels in the dorsal hippocampus of female mice, an area with the highest GR expression [34], which plays a crucial role in the feedback regulation of the HPA axis [61]. The majority of GRs are nuclear receptors that directly bind CORT and elicit either gene transcription or gene repression via binding to glucocorticoid receptor elements (GRE) on DNA [62]. Overstimulation of this receptor by extremely high CORT levels may have induced downregulation of GR, most likely through the ubiquitin proteasome pathway [63] to regulate transcription levels.

Interestingly, we also observed a female-specific reduction of GR in HRM versus WT controls, reflecting that reelin deficiency leads to reduced GR expression in the dorsal hippocampus in females only. GR is expressed in almost every cell type throughout the body and brain [64,65] with high abundancy in the hippocampus and PFC [34,66]. Extensive evidence shows that reelin promotes dendrite and spine formation during early development, particularly in the hippocampus and the cortex [9,15]. Korn et al. [67] recently showed that abnormal reelin signaling decreased neurogenesis and increased the number of hilar ectopic dentate granule cells in adult mice. According to the GeneMANIA prediction server [68] as well as the literature there is no evidence for a direct physical interaction between reelin and GR receptors, hence the GR reduction found here is most likely the result of an indirect or compensatory response to reelin deficiency. Lower GR levels in female reelin-deficient mice may also be due to lower numbers of neurons and dendrites in these animals. Further studies are needed to investigate whether there are sex differences in neuronal expression, dendrite formation and what type of cells are affected specifically in reelin haploinsufficient models. However, prenatal treatment with the corticosteroid dexamethasone was previously shown to reduce hippocampal calretinin expression in female but not male rats [69], suggesting a female-specific vulnerability of these cells that are known to co-express reelin [70]. Our finding of female-specific susceptibility to reelin deficiency is supported by studies showing that a single nucleotide polymorphism (SNP) with the *RELN* gene sequence increases the risk of schizophrenia in women, but not in men [71,72]. In this study we show an additive effect of reelin deficiency and CORT treatment on GR expression in female mice.

We further showed that late-adolescent CORT treatment reduced GluN2B and GluN2C subunits of the NMDA receptor in the dorsal hippocampus and GluN1 in the PFC of female but not male mice. We [24] previously reported that CORT reduced GluN2C protein levels in male mice in the PFC and dorsal hippocampus independent of the genotype. We did not detect this change in this study, which may be explained by the different CORT dose used (25 mg/L vs. 50 mg/L in the present study). The direct effect of CORT on NMDA receptors is supported by an in vitro study showing that CORT application reduces calcium influx through NMDA receptors in hippocampal slices [73]. Reelin also modulates calcium influx through the NMDA receptors, however our study shows that while CORT also modifies GluN1 receptor expression, reelin haploinsufficiency does not. Zhang et al. [74] further demonstrated that glucocorticoids suppress NMDAr activity by acting on putative non-genomic G-protein-coupled receptors activating the phospholipase C (PLC) pathway. Hence, GluN2B downregulation may result from a direct interaction with the receptor through the PLC pathway or may be subsequent to changes in other molecules. Once again, the female-specific CORT effect on NMDAr subunits may be due to the 3-fold higher CORT levels as compared to males. Van den Buuse et al. [75] previously reported significant up-regulation of NR1 subunits, but down-regulation of NR2C subunits in PFC of male and female HRM, while we could not detect these differences in the current study. This may be due to the fact that van den Buuse et al. [75] collected the brains starting at 12 weeks of age, while we looked at the brains from 15 week old mice. Furthermore, while we separated male and female groups, van den Buuse et al. [75] combined the two sexes for statistical analysis.

Reelin has been implicated in the regulation of PV and GAD_67_ expression [23,76,77,78], however we did not find any differences in PV or GAD_67_ levels in any of the three brain areas. Supporting our results, Lussier at al. [79] did not detect changes in hippocampal GAD_67_ expression in HRM. By contrast, as opposed to our findings, Costa et al. [78] showed depletion of GAD_67_ in the frontal cortex of HRM and Nullmeier et al. [23] demonstrated reduced GAD_67_ and PV in the hippocampus of HRM. The contradicting results may be due to different techniques and brain regions used for analysis. While Nullmeier et al. [23] compared sub-regions of the hippocampus such as CA1, CA2 or dentate gyrus by means of immunohistochemistry, we performed Western blots on ventral and dorsal hippocampi. Nullmeier et al. [23] analyzed male and female groups together, while other studies that are mentioned used only males for detection of PV and GAD_67_, which may explain contradicting results. Differences in age of the analyzed brain as well as housing conditions may also contribute to inconsistent results in the literature. While early-life stress and chronic isolation stress were shown to reduce PV in the hippocampus [80], pharmacological stress during late-adolescence in our study did not affect PV or GAD_67_. The former study, however, did not look at the long-term effects of chronic stress, but obtained tissue samples immediately at the end of the stress paradigm, whereas our study included 6 weeks of washout period after treatment cessation. It is possible that in our animals, some early CORT-induced changes in molecular markers could have been restored by this time. Therefore, different types of stress and different periods of stress application may have differential effects on these proteins. Furthermore, cellular distribution of these proteins may have been overlooked and immunohistochemical analysis would be of relevance to confirm these results. Indeed, a limitation of this study is the lack of cellular specificity afforded by the use of tissue homogenates for Western blot analysis. Future studies using immunohistochemistry techniques to identify cell-specific alterations in GR expression, as well as GABAergic markers in this two-hit model would be advantageous. In addition, an important caveat to this study is that the brain tissue was removed 6 weeks following the chronic corticosterone treatment, with the intention to assess the long term effects of adolescent exposure to corticosterone. Here a parallel group, whereby tissue was collected immediately following or during corticosterone treatment would be a good comparison to compare immediate and long term effects of corticosterone on these protein targets.

In conclusion, CORT reduced NMDAr subunit expression in the dorsal hippocampus and PFC in female, but not male mice. A female specific reduction of GR in response to CORT was also observed in the dorsal hippocampus, which may be explained by the 3-fold higher CORT levels in females compared to male mice during the treatment period. In addition, GR expression was decreased in female HRM. Albeit reelin depletion did not enhance the effects of CORT on protein expression of NMDAr subunits, GR, GAD_67_ or PV as measured by Western blot analysis, an additive effect of reelin depletion and CORT on GR expression was observed in female but not male dorsal hippocampus. This is an important direction for future studies as reelin deficiency as well as stress and related HPA–axis dysregulation play a major role in the pathophysiology of numerous mental disorders, such as MDD or schizophrenia. In addition, a major advantage of this study was the incorporation of sex as a biological variate. Here we found distinct changes in female mice that may not have been uncovered if both sexes were combined, and that provide important direction for future studies investigating the potential role of sex steroid hormones in modulating reelin and stress-induced molecular changes.

## Figures and Tables

**Figure 1 brainsci-08-00118-f001:**
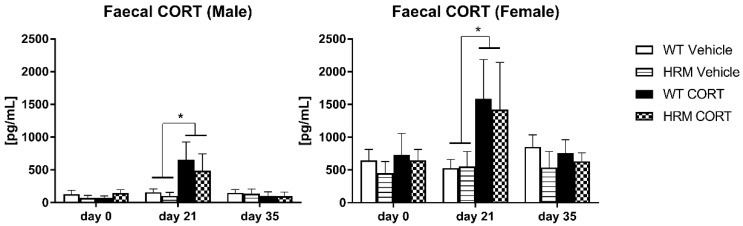
CORT levels in faecal boli at day 0 (before start of CORT treatment), day 21 (last day of CORT treatment) and day 35 (2 weeks after cessation of CORT treatment) in WT vehicle-treated mice (white bars), HRM vehicle-treated mice (lined bars), WT CORT-treated mice (black bars) and HRM CORT-treated mice (checked bars) Data are expressed as mean ± SEM (*n* = 4–6). A main treatment effect was observed on day 21 with higher CORT levels in CORT-treated groups (black and checked bars) compared to controls (* *p* < 0.05). Female mice showed significantly higher CORT levels than male mice at all three time points of measurement. No treatment or genotype differences were detected on days 0 or day 35. Heterozygous reelin mice (HRM); wildtype (WT).

**Figure 2 brainsci-08-00118-f002:**
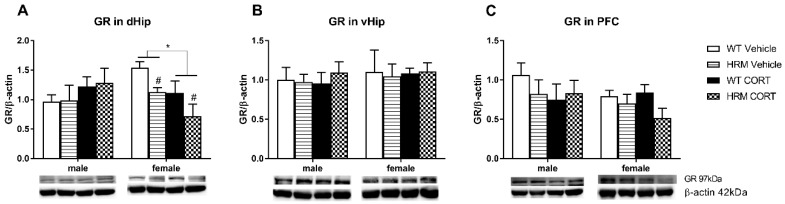
Glucocorticoid receptor (GR) protein expression in male and female WT vehicle-treated mice (white bars), HRM vehicle-treated mice (lined bars), WT CORT-treated mice (black bars) and HRM CORT-treated mice (checked bars). (**A**) GR levels were significantly lower in the dHip of HRM female, but not male mice compared to WT controls (^#^
*p* < 0.05). In addition CORT significantly reduced GR expression in the dHip of female mice (* *p* < 0.05). (**B**) GR levels were unchanged in the ventral hippocampus (vHip) and prefrontal cortex (PFC) (**C**). Data are expressed as mean ± SEM (*n* = 5–6). Heterozygous reelin mice (HRM); wildtype (WT).

**Figure 3 brainsci-08-00118-f003:**
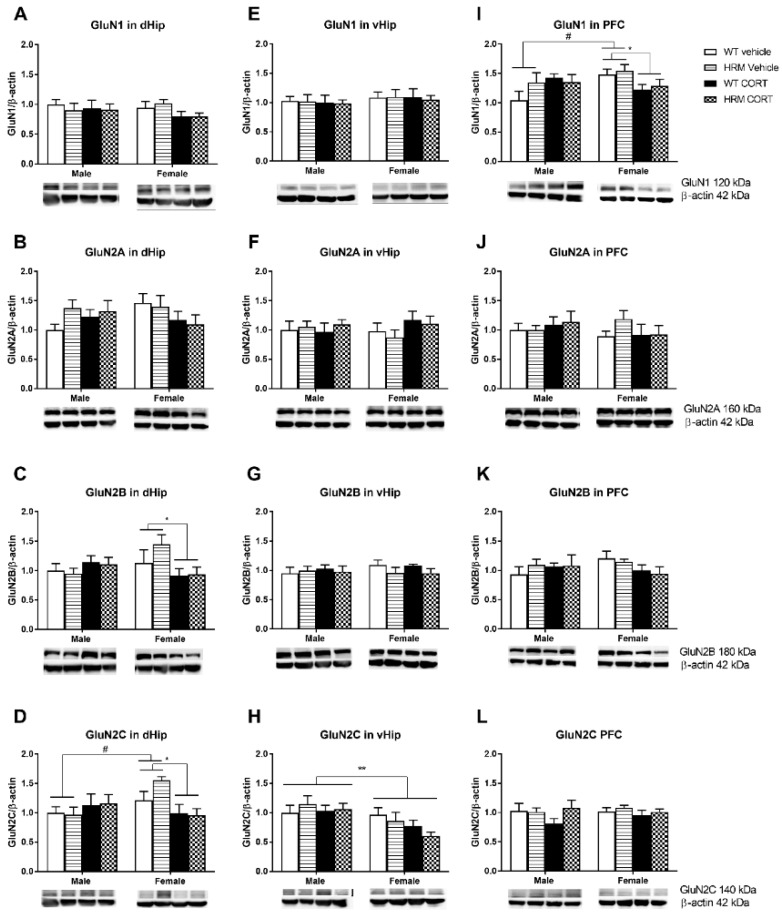
Protein expression of NMDAr subunits GluN1(**A**,**E**,**I**), GluN2A (**B**,**F**,**J**), GluN2B (**C**,**G**,**K**) and GluN2C (**D**,**H**,**L**) in the dorsal hippocampus (dHip) (**A**–**D**), ventral hippocampus (vHip) (**E**–**H**) and prefrontal cortex (PFC) (**I**–**L**). Data are represented as mean ± SEM (*n* = 4–6). Analysis revealed no differences in GluN1 (**A**) or GluNRA (**B**) in the dHip. However, a female-specific reduction in GluN2B (**C**) and GluN2C (**D**) was found in response to CORT in the dHip (* *p* < 0.05), and in vehicle-treated animals GluN2C levels (**D**) were higher in female mice compared to male mice (^#^
*p* < 0.05). No differences were detected between the experimental groups in the vHip (**E**–**H**), although female mice had lower levels of GluN2C (**H**) compared to males (** *p* < 0.01). In the PFC CORT reduced GluN1 (**I**) in female mice irrespective of the genotype (* *p* < 0.05), while in vehicle-treated animals GluN1 was higher in female mice compared to male mice (^#^
*p* < 0.05). No significant differences were seen in GluN2A (**J**), GluN2B (**K**) or GluN2C (**L**) levels in PFC between the groups. Heterozygous reelin mice (HRM); wildtype (WT).

**Figure 4 brainsci-08-00118-f004:**
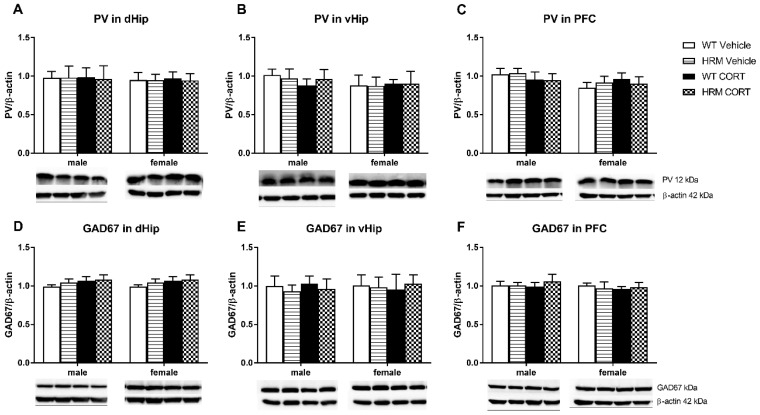
Levels of PV were unchanged between the groups in the dHip (**A**), vHip (**B**) and PFC (**C**). Protein levels of GAD_67_ were unchanged in the dorsal hippocampus (dHip) (**D**), ventral hippocampus (vHip) (**E**) and prefrontal cortex (PFC) (**F**). Data are represented as mean ± SEM (*n* = 5–6). Heterozygous reelin mice (HRM); wildtype (WT).

**Table 1 brainsci-08-00118-t001:** Relative adrenal weight.

	Male	Female
WT Contr	1.13 ± 0.05	2.10 ± 0.11 ^#^
HRM Contr	1.13 ± 0.09	1.97 ± 0.09 ^#^
WT CORT	0.92 ± 0.02 *	1.85 ± 0.09 *^,#^
HRM CORT	0.96 ± 0.08 *	1.82 ± 0.08 *^,#^

Data are represented as mean ± SEM of *n* = 8–16; Relative adrenal weight was calculated as adrenal weight/body weight × 10,000; * *p* < 0.05 for difference between CORT groups and controls as shown by ANOVA main effect; ^#^
*p* < 0.05 for significant sex difference as shown by ANOVA main effect; heterozygous reelin mice (HRM); wildtype (WT).

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
