# Peer review of "Reelin Haploinsufficiency and Late-Adolescent Corticosterone Treatment Induce Long-Lasting and Female-Specific Molecular Changes in the Dorsal Hippocampus"

_brainsci, 2018, doi:10.3390/brainsci8070118_

Round 1

Reviewer 1 Report

This is, in summary, an interesting manuscript aimed to explore the interaction between reelin depletion and stress in male and female heterozygous reelin mice (HRM) and wildtype (WT) controls treated with the stress hormone, and corticosterone (CORT) during late adolescence. The authors found that, while no changes were observed in male mice, female HRM showed a significant reduction in GR expression in the dorsal hippocampus. Moreover, CORT reduced GR levels together with GluN2B and GluN2C subunits of NMDAr in the dorsal hippocampus in female mice only. Furthermore, CORT reduced GluN1 levels in the PFC of female mice and the combined effect of HRM and CORT treatment appeared to be additive in terms of GR expression in the dorsal hippocampus. In addition, female-specific CORT-induced changes were associated with overall higher circulating CORT levels in female compared to male mice.

The authors may find as follows my main comments/suggestions.

First, as the authors, throughout the Introduction section, stressed the importance of the correct position of neurons throughout the hippocampus as well as the role of reelin in psychiatric disorders, they could also mention the possible association between impaired neuroplasticity mechanisms, neurotrophic factors, and some neuropsychiatric conditions such as major depression. In particular, either chronic stress and depressive symptoms are associated with structural brain changes such as a loss of dendritic spines and synapses as well as reduced dendritic arborisation together with reduced glial cells in the hippocampus. Although i understand that the association between impaired neurogenesis, neurotrophic factors, and specific neuropsychiatric conditions is not the main topic of the present paper, the authors might refer, at least briefly, to the specified topic and their implications in terms of neurotoxicity and its consequences (e.g., brain demage) in the mentioned brain areas. Thus, in order to focus on this issue, i suggest to cite within the main text the paper published on CNS & Neurological Disorders - Drug Targets in 2014 (PMID: 25470403). This is particularly relevant as the same authors, more ahead (within the Discussion section) focused on the assumption that extensive evidence shows that reelin promotes dendrite and spine formation during early development, particularly in the hippocampus and the cortex.

In addition, within the same section some statements such as “findings on the effect of reelin deficiency on the expression of GABAergic markers as well as NMDAr subunits remain inconsistent, possibly due to a lack of stratification for sex “ or “given that a combination of genetic predisposition and environmental insults such as stress contributes to psychiatric illnesses, reelin deficiency may constitute a risk factor by making the brain more vulnerable to stress” are interesting as presented but need to be better developed and supported by adequate refences for the general readership.

Moreover, considering that the main aims of this paper have been extensively proposed by the authors, the specific hypotheses underlying the study objectives could be adequately reported.

Furthermore, the authors do not need to repeat within the first lines of the Discussionse section the main main of this study (identify whether a genetic deficit in reelin increases susceptibility to high glucocorticoid levels during late adolescence as measured by relevant molecular changes in the hippocampus and PFC of male and female mice) as this objective has been already well described elsewhere. Thus, i suggest to immediately focus on the main findings of the manuscript and its most relevant implications for the general readership.

Also, what the authors may suggest about the possible existence of sex differences in neuronal expression, dendrite formation, and type of cells which are affected specifically in reelin haploinsufficient models? While the authors simply state that further studies are needed to investigate this complex theme, i believe that they could try to provide, at least briefly, their specific point of view to this regard according to their expertise.

Finally but most importantly, the authors should more deeply stress the most relevant limitations/shortcomings of this study in order to provide a more critical perspective of this paper to the readers.

Author Response

Reviewer 2.

The authors may find as follows my main comments/suggestions.

First, as the authors, throughout the Introduction section, stressed the importance of the correct position of neurons throughout the hippocampus as well as the role of reelin in psychiatric disorders, they could also mention the possible association between impaired neuroplasticity mechanisms, neurotrophic factors, and some neuropsychiatric conditions such as major depression. In particular, either chronic stress and depressive symptoms are associated with structural brain changes such as a loss of dendritic spines and synapses as well as reduced dendritic arborisation together with reduced glial cells in the hippocampus. Although i understand that the association between impaired neurogenesis, neurotrophic factors, and specific neuropsychiatric conditions is not the main topic of the present paper, the authors might refer, at least briefly, to the specified topic and their implications in terms of neurotoxicity and its consequences (e.g., brain demage) in the mentioned brain areas. Thus, in order to focus on this issue, i suggest to cite within the main text the paper published on CNS & Neurological Disorders - Drug Targets in 2014 (PMID: 25470403). This is particularly relevant as the same authors, more ahead (within the Discussion section) focused on the assumption that extensive evidence shows that reelin promotes dendrite and spine formation during early development, particularly in the hippocampus and the cortex.

The following has been added to the introduction, line 52:

In a comprehensive review of articles published from 1980-2014 on major depression a main finding was the association between both chronic stress and major depression with dendritic spine loss and aborization as well as reduced glial cells [17], thus disruption to reelin may be upstream of this broad set of functional deficits associated with major depression.   

In addition, within the same section some statements such as “findings on the effect of reelin deficiency on the expression of GABAergic markers as well as NMDAr subunits remain inconsistent, possibly due to a lack of stratification for sex “ or “given that a combination of genetic predisposition and environmental insults such as stress contributes to psychiatric illnesses, reelin deficiency may constitute a risk factor by making the brain more vulnerable to stress” are interesting as presented but need to be better developed and supported by adequate references for the general readership.

We have expanded on this statement adding the following, line 72:

Nullmeier et al. [22] for example combined male and female data when comparing GAD67 expression in reelin haploinsufficient and wild-type mice and found a significant decrease in GAD67 and PV in multiple hippocampal regions, but did not stratify for sex in the analysis. Previously, we have shown sex differences in BDNF, TrkB and GluNR2C expression in reelin haploinsufficient mice,  [24,25], while others have shown that male but not female Rln+/- mice show alterations in steroid hormone levels in the cerebellum when compared to controls [26].

“given that a combination of genetic predisposition and environmental insults such as stress contributes to psychiatric illnesses, reelin deficiency may constitute a risk factor by making the brain more vulnerable to stress”.

This has been changed on line 81 to:

While several genetic variants have been associated with severe psychiatric disorders such as major depression, no single gene is likely to be causative. Rather a gene x gene or gene x environment interaction is more probable. A well-accepted model to explain the complex and heterogeneous aetiology of psychiatric disorders is the two-hit hypothesis, which suggests that early genetic or environmental insults disrupt the developing brain causing it to be vulnerable to a second environmental insult during adolescence, such as stress [30]. Reelin deficiency may constitute a risk factor by…

Moreover, considering that the main aims of this paper have been extensively proposed by the authors, the specific hypotheses underlying the study objectives could be adequately reported.

The following has been added to line 111:

We hypothesize that the combined effect of reelin haploinsufficiency and adolescent CORT treatment will cause alterations in GABAergic, glutamatergic and stress receptor pathways but these may be differentially modulated according to sex.

Furthermore, the authors do not need to repeat within the first lines of the Discussionse section the main main of this study (identify whether a genetic deficit in reelin increases susceptibility to high glucocorticoid levels during late adolescence as measured by relevant molecular changes in the hippocampus and PFC of male and female mice) as this objective has been already well described elsewhere. Thus, i suggest to immediately focus on the main findings of the manuscript and its most relevant implications for the general readership.

This has been deleted

Also, what the authors may suggest about the possible existence of sex differences in neuronal expression, dendrite formation, and type of cells which are affected specifically in reelin haploinsufficient models? While the authors simply state that further studies are needed to investigate this complex theme, i believe that they could try to provide, at least briefly, their specific point of view to this regard according to their expertise.

The following has been added to the discussion, line 340:

However, prenatal treatment with the corticosteroid dexamethasone was previously shown to reduce hippocampal calretinin expression in female but not male rats [70], suggesting a female-specific vulnerability of these cells that are known to co-express reelin [71].

Finally but most importantly, the authors should more deeply stress the most relevant limitations/shortcomings of this study in order to provide a more critical perspective of this paper to the readers.

The following has been added, line 388 discussion:

Indeed, a limitation of this study is the lack of cellular specificity afforded by the use of tissue homogenates for Western blot analysis.

Line 391: In addition, an important caveat to this study is that the brain tissue was removed 6 weeks following the chronic corticosterone treatment, with the intention to assess the long term effects of adolescent exposure to corticosterone. Here a parallel group, whereby tissue was collected immediately following or during corticosterone treatment would be a good comparison to compare immediate and long term effects of corticosterone on these protein targets.

Reviewer 2 Report

1.     All of the figure legends are too simple to understand the figures.

2.     No advantages and disadvantages of current study in the result or conclusion

section.

3.     No future directions.

4.     Some typos, grammatical errors and format issues.

Author Response

1.                  All of the figure legends are too simple to understand the figures.

All figure legends have been adjusted.

2.     No advantages and disadvantages of current study in the result or conclusion 

section.

The following has been added to line 394:

In addition, a major advantage of this study was the incorporation of sex as a biological variate. Here we found distinct changes in female mice that may not have been uncovered if both sexes were combined, and that provide important direction for future studies investigating the potential role of sex steroid hormones in modulating reelin and stress-induced molecular changes.

Limitations of the study have been included, line 376:

Line 376: Indeed, a limitation of this study is the lack of cellular specificity afforded by the use of tissue homogenates for Western blot analysis.

Line 379: In addition, an important caveat to this study is that the brain tissue was removed 6 weeks following the chronic corticosterone treatment, with the intention to assess the long term effects of adolescent exposure to corticosterone. Here a parallel group, whereby tissue was collected immediately following or during corticosterone treatment would be a good comparison to compare immediate and long term effects of corticosterone on these protein targets.

3.      No future directions.

The following has been added:

Line 377: Future studies using immunohistochemistry techniques to identify cell-specific alterations in GR expression, as well as GABAergic markers in this two-hit model would be advantageous.

Line 381: Here a parallel group, whereby tissue was collected immediately following or during corticosterone treatment would be a good comparison to compare immediate and long term effects of corticosterone on these protein targets.

4.      Some typos, grammatical errors and format issues.

We have made every effort to correct grammatical or format issues. Are there any specific examples of errors?

Round 2

Reviewer 1 Report

In the revised manuscript, the authors have now addressed most of the major questions raised by Reviewers improving the main structure and quality of the paper. I have no further additional comments.

Reviewer 2 Report

 I accept the revised version of the manuscript.